# Low-Luminance Blue Light-Enhanced Phototoxicity in A2E-Laden RPE Cell Cultures and Rats

**DOI:** 10.3390/ijms20071799

**Published:** 2019-04-11

**Authors:** Cheng-Hui Lin, Man-Ru Wu, Wei-Jan Huang, Diana Shu-Lian Chow, George Hsiao, Yu-Wen Cheng

**Affiliations:** 1School of Pharmacy, College of Pharmacy, Taipei Medical University, Taipei 11031, Taiwan; d301097008@tmu.edu.tw (C.-H.L.); eel2243@tmu.edu.tw (M.-R.W.); 2Graduate Institute of Pharmacognosy, Taipei Medical University, Taipei 11031, Taiwan; wjhuang@tmu.edu.tw; 3Institute of Drug Education and Research, College of Pharmacy, University of Houston, Texas 77004, USA; DChow@uh.edu; 4Department of Pharmacology, School of Medicine, College of Medicine, Taipei Medical University, Taipei 11031, Taiwan; 5Graduate Institute of Medical Sciences, College of Medicine, Taipei Medical University, Taipei 11031, Taiwan

**Keywords:** blue light, N-retinylidene-N-retinylethanolamine (A2E), retinal pigment epithelial cells (RPE), retinal damage

## Abstract

N-retinylidene-N-retinylethanolamine (A2E) and other bisretinoids are components of lipofuscin and accumulate in retinal pigment epithelial (RPE) cells—these adducts are recognized in the pathogenesis of retinal degeneration. Further, blue light-emitting diode (LED) light (BLL)-induced retinal toxicity plays an important role in retinal degeneration. Here, we demonstrate that low-luminance BLL enhances phototoxicity in A2E-laden RPE cells and rats. RPE cells were subjected to synthetic A2E, and the effects of BLL on activation of apoptotic biomarkers were examined by measuring the levels of cleaved caspase-3. BLL modulates the protein expression of zonula-occludens 1 (ZO-1) and paracellular permeability in A2E-laden RPE cells. Early inflammatory and angiogenic genes were also screened after short-term BLL exposure. In this study, we developed a rat model for A2E treatment with or without BLL exposure for 21 days. BLL exposure caused fundus damage, decreased total retinal thickness, and caused neuron transduction injury in the retina, which were consistent with the in vitro data. We suggest that the synergistic effects of BLL and A2E accumulation in the retina increase the risk of retinal degeneration. These outcomes help elucidate the associations between BLL/A2E and angiogenic/apoptotic mechanisms, as well as furthering therapeutic strategies.

## 1. Introduction 

Abnormal regeneration of the visual retinoid (retinaldehyde) chromophores—between photoreceptor cells and retinal pigment epithelial (RPE) cells—non-enzymatically forms lipofuscin, thereby threatening the survival of retinal cells. These chromophores are involved in the pathogenesis of retinal degenerative diseases such as age-related macular degeneration (AMD) and Stargardt’s disease (STGD) [1,2]. Dry (atrophic) AMD accounts for most AMD cases, and is characterized by RPE degeneration and damage followed by atrophy and diminished nearby photoreceptors, causing retinal thinning and even vision loss [3]. Although wet (neovascular) AMD can be treated with several anti-angiogenic drugs via intravitreous administration, the etiology of dry-AMD remains uncertain because of multiple pathological factors, including continuous oxidative stress and chronic inflammation [4]. 

RPE cells form an epithelial monolayer located in the retina and have many functions, including absorption of light, ionic exchange, visual cycle, transportation of nutrients and wastes, phagocytosis of shed photoreceptors, as well as metabolism and immune responses [5]. Owing to a high metabolic rate, RPE cells are characterized as being susceptible to photo-oxidative injury [6]. Initial dysfunction and degeneration of RPE cells usually result in the accumulation of metabolic waste (i.e., lipofuscin) between choroidal and RPE layers. Given that RPE cells play an important role in the progression of AMD, the close association between photoreceptors and RPE cells should be emphasized. As such, RPE injury and cell death leads to secondary degeneration of photoreceptors, which characterizes advanced dry-AMD, also known as geographic atrophy [7,8].

Drusen contain mixtures of lipofuscin that are very photo-sensitive, and are involved in the production of singlet oxygen and superoxide anion, thus initiating photo-sensitive reactions [9,10]. A2E is a major component of drusen, with a maximum wavelength absorbance of 460 nm. Direct transmittal of light to the retina via drusen results in cleavage of A2E at the pyridinium ring, leading to oxidative stress [11]. Blue light-emitting diode (LED) light (BLL)-irradiation further oxidizes A2E and its derivatives, and enhances RPE cell dysfunction, including differentiation, inflammation, and apoptosis, attributable to photo-oxidation [12,13]. Excessive light exposure induces oxidative stress in photoreceptors/RPE cells, causing abnormal inflammation, angiogenesis, and apoptosis [14]. Moreover, in primary human RPE cells, cumulative light exposure causes decremental Bcl-2 expression and incremental Bax and vascular endothelial growth factor (VEGF) expression, which promote the development of wet-AMD [15]. In addition, high-luminance light results in deformation and disruption of RPE junction proteins such as zonula-occludens 1 (ZO-1), and immediately impacts the outer blood–retinal barrier (BRB) [16]. 

In this study, we show that co-treatment with BLL and synthetic A2E up-regulates apoptotic proteins such as cleaved caspase-3 in RPE cells. Further, BLL down-regulates expression of tight junction proteins such ZO-1, increases the expression of angiogenic genes, and increases paracellular permeability in A2E-laden RPE cells. Using a long-term low-luminance periodic BLL exposure model in brown Norway (BN) rats, we demonstrate that co-treatment with A2E and BLL causes more severe retinal damage and neuronal transmission dysfunction than A2E treatment alone. This manuscript provides further insight into the pathophysiology associated with the combination of BLL exposure and A2E accumulation, as well as information regarding a new strategy for the treatment of macular degeneration.

## 2. Results

### 2.1. A2E Damages RPE Cells under Blue Light Exposure

To investigate A2E-induced toxicity in RPE cells, we applied synthetic A2E to RPE cells. The MTT assay was used to assess viability of cultured RPE cells in response to different concentrations (0, 10, 20, 30, 40, 60, 80, and 100 µM) of A2E. Viability decreased in cells treated with 60, 80, and 100 µM A2E (Figure 1A, white bars). Moreover, in cells co-treated with BLL and A2E for 12 h, a reduction in cell viability was observed in an A2E dose-dependent manner. Cell viability was decreased by 50% in A2E-treated (30 µM) cells exposed to BLL for 12 h, demonstrating that BLL increases A2E toxicity (Figure 1A, red bars). A concentration of 30 µM A2E was chosen for further experiments to investigate BLL-induced toxicity in the presence of A2E. Cell morphology was also observed via microscopy. The data showed that co-treatment of RPE cells with 30 µM A2E and BLL resulted in a slim and shrinking morphology (Figure 1Bd). Confocal microscopy revealed a dose-dependent increase in autofluorescence in the cytoplasm of A2E-laden RPE cells, as well as pyknotic and fragmented nuclei (Figure 1Ce,h,k). Immunofluorescent staining showed apoptosis of RPE cells, attributable to cleavage of caspase-3 (Figure 2). A2E resulted in upregulation of caspase-3 cleavage forms (Figure 2e), and caused nuclei damage (Figure 2d). Moreover, co-treatment with A2E and BLL increased the number of cleaved caspase-3 forms moving into the nuclei of A2E-laden RPE cells and consistently caused apoptosis (Figure 2k,l). These data are likely attributable to A2E-induced cytotoxicity, which increased after BLL exposure, owing to up-regulation of cleaved caspase-3 in RPE cells.

### 2.2. A2E Reduces Tight Junction Protein Expression and Increases Paracellular Permeability in RPE Cells

To examine how synthetic A2E affects RPE cells, several experiments were executed and sequential exposure to A2E was tested. First, we evaluated the cellular integrity of RPE cells treated with A2E for different concentrations (0, 5, 10, and 30 µM) and times (3, 6, 12, 24, and 48 h) via the trans-epithelial electrical resistance (TEER) assay, which represents an early expression of cell damage when reductions in TEER are observed [17]. Normalized TEER values in RPE cells decreased from 86.26 ± 4.70 to 62.60 ± 2.86% after treatment with A2E for 12 h. TEER was significantly lower in cells treated with A2E compared to controls (Figure 3A). Furthermore, we added 30 µM A2E to RPE cells under different times of BLL exposure, and found that TEER was significantly lower in the A2E + BLL co-treated group (Figure 3B). Next, we measured ZO-1 via immunofluorescence staining in A2E-laden RPE cells exposed to BLL to evaluate the expression and integrity of tight junction proteins. ZO-1 is a subtype from the tight junction protein family and functions as a cross-linker, playing an important role in the blood–retinal barrier [18]. ZO-1-positive strands were noted in cells treated with 30 µM A2E for 12 h. However, intact ZO-1-positive strands were still observable (Figure 3Cg, yellow arrows). In contrast, immunofluorescence staining showed an increase in discontinued or intermittent ZO-1-positive strands displaying intercellular gaps in A2E-laden RPE cells exposed to BLL for 12 h (Figure 3Co, yellow arrows), indicating that the tight junction structure was damaged. The above data suggest that BLL induces early damage in A2E-laden RPE cells by modulating tight junction protein expression.

### 2.3. Blue Light Induces Early Inflammation and Angiogenesis in RPE Cells and Increases VEGF Levels in A2E-Laden RPE Cells 

In addition to reduced TEER and disintegration of tight junction proteins in A2E-laden RPE cells, we further examined whether BLL affects genes involved in inflammation (*CFH*, *IL-1β*, and *IL-12*) and angiogenesis (*Ang-1*, *CTGF, CCL-2*, *EPO, GLUT-1, MMP-9*, *OPTC*, and *VEGFR-2*), which we measured via quantitative PCR (Figure 4A). Expression of mRNAs was higher for several inflammation- and angiogenesis-related genes in RPE cells exposed to BLL for 2 h than that in controls; specifically, *CCL-2, GLUT-1, IL-1β*, and *MMP-9* were upregulated. In contrast, mRNA expression of *EPO* was downregulated after BLL exposure (Figure 4B). We next examined the level of VEGF secreted by RPE cells by means of an enzyme-linked immunosorbent assay (ELISA). The concentration of VEGF increased from 1145.62 ± 103.80 to 2101.83 ± 172.67 pg/mL after treatment with BLL for 12 h in the A2E-laden RPE cells (Figure 4C). These results demonstrate that BLL increases inflammation and angiogenesis in A2E-laden RPE cells.

### 2.4. Blue Light Reduces Retinal Thickness and Damages Retinal Neuron Transduction in BN Rats

To further evaluate the photo-toxic effects of BLL, we subjected BN rats to BLL exposure cages that were designed for animals. Our in vivo experimental design is presented in Figure 5A, andBrown Norway (BN) rats were divided into four group: control group, BLL-exposed group, A2E-treated group, and A2E + BLL co-treated group. BN rats were subjected to the examinations on day 0, day 7, day 14, and day 21 (star marks) including bright field (BF), fundus fluorescein angiography (FFA), spectral-domain optical coherence tomography (SD-OCT), and electroretinographic (ERG) recordings. Before the periodic BLL exposure, BN rats were dark-adapted on day -7 to eliminate the effects of light exposure from their previous environment. After 7 days of dark adaption, we first collected ERG recordings, then the rats were daily exposed to BLL for 3 h, followed by a resting period of 21 h on day 0. On day 7, day 14, and day 21 (Figure 5A, star marks), we subjected BN rats to the animal ophthalmoscope to obtain fundus images in the bright field, fluorescein angiography and Spectral-Domain Optical Coherence Tomography (SD-OCT)scanning. We did not observe any damage in the control group, as determined from the bright-field images, sodium angiography, and SD-OCT scanning (Figure 5Ba-l). However, we observed blurring and a white area in the bright field in BN rats exposed to periodic BLL for 3 h per day for 7 days (Figure 5Cd). On days 14–21, the white area turned into a clear small white dot with well-defined margins (Figure 5Cg), and became larger on day 21 (Figure 5Cj). Further, periodic BLL exposure induced vessel tortuosity on day 7 (Figure 5Ce) and day 14 (Figure 5Ch), as shown by fluorescein angiography. On day 21, fluorescein leakage was observed in the shape of a half-moon on the upper-right side of the retina (Figure 5Ck). In addition, SD-OCT results clearly showed that periodic BLL exposure induces retinal atrophy (Figure 5Cc,f,i,l). From day 7 to 21, choroidal vessel diameter and thickness were further enlarged after periodic BLL exposure, resulting in damage to the photoreceptors and RPE cells as observed via SD-OCT scanning (Figure 5l). The analytic data reveal that total retinal thickness and inner segment/outer segment (IS/OS)-RPE layers were significantly decreased after 21 days of periodic BLL exposure (Figure 5D,G). Though retinal vessel tortuosity was observed after periodic BLL exposure on days 7 and 14, no changes in retinal thickness were observed between treated and control rats. Therefore, we investigated the effects of periodic BLL exposure on retinal neuron transduction, which is determined by changes in potassium and sodium ions. By examining ERG recordings, we evaluated the transduction ability of neurons in BN rats periodically exposed to BLL. The a- and b-waves were lower in rats exposed to 21 days of periodic BLL than those in the control group (Figure 7A), as shown in the electrooculogram. Further, periodic BLL exposure significantly reduced a- and b-wave amplitudes on day 7, day 14 and 21 (Figure 7D,E), without changes in implicit time (Figure 7B,C). 

### 2.5. Blue Light Enhances Retinal Neuron Transduction Injury in A2E-Treated BN Rats

Next, we applied synthetic A2E (2 µL, 30 µM) to the right eye of BN rats via intravitreal injection. A2E induced retinal vessel tortuosity on day 7 (Figure 6Ae), which was even more pronounced on days 14 and 21 (Figure 6Ah,Ak). However, bright field fundus images and retinal thickness (as shown by SD-OCT) remained unchanged. Interestingly, retinal damage was observed on day 7 (Figure 6Bd) in rats co-treated with A2E and periodic BLL, worsening by days 14 and 21, as observed in the fundus images (Figure 6Bg,j). Though no obvious retinal angiogenesis was found via fluorescein angiography, several minor vessel leakages were observed on day 7 (Figure 6Be), and severe fluorescein leakage in the shape of a half-moon was observed on the upper-right side of the retina (Figure 6Bh,k). Moreover, SD-OCT clearly showed retinal atrophy and a diminishing photoreceptor layer in rats co-treated with A2E and periodic BLL (Figure 6Bi,l). From days 7 to 21, choroidal vessel diameter and thickness were further enlarged after periodic BLL exposure, resulting in damage to the photoreceptors and RPE cells as observed via SD-OCT scanning. The analytic data show that co-treatment with A2E and periodic BLL exposure results in significant decrements in total retinal thickness, inner plexiform layer (IPL)-IS/OS, and IS/OS-RPE cells; especially, co-treatment with A2E increased periodic BLL exposure-induced toxicity to RPE cells in BN rats (Figure 6C,E,F). Next, we evaluated the effects of co-treatment with A2E and periodic BLL on neuron transduction in BN rats. Electroretinography (ERG) a- and b-waves decreased in rats co-treated with A2E and periodic BLL for 21 days, as shown in the electrooculogram (Figure 7A). Further, co-treatment with A2E and periodic BLL significantly reduced a- and b-wave amplitudes on days 7, 14, and 21 (Figure 7D,E). In addition, A2E and periodic BLL exposure prolonged the implicit time of the b-wave on day 21, providing evidence of neuron transduction degeneration in the retina (Figure 7C). These data provide evidence that periodic BLL exposure damages retinas in combination with A2E treatment in the early stages and decreases retinal neuron transduction, potentially affecting the visual sense. 

## 3. Discussion

A2E is a major hydrophobic component of lipofuscin [19]. Various analogs in between the photoreceptor OS and apical membrane of RPE cells can be generated during the multiple steps involved in the visual cycle, including A2-glycero-phosphoethanolamine (A2-GPE), A2-dihydropyridine-phosphatidylethanolamine (A2-DHP-PE), and all-*trans*-retinal dimer phosphatidylethanolamine (all-*trans*-retinal dimer-PE) [20]. Massive accumulation of photosensitive A2E and other all-*trans*-retinal dimers leads to the formation of lipofuscin/drusen, which are located between the choroidal and RPE layers. Choroidal capillaries are barely able to remove these heterogeneous debris-containing waste products, gradually leading to inflammation, neovascularization, and even hemorrhage [21,22]. To further this knowledge, in this study we evaluated pathological responses, including inflammation, angiogenesis, and apoptosis by assessing the effects of synthetic A2E and BLL exposure to RPE cells and BN rats. A2E was primarily distributed to the cytoplasm when RPE cells were treated with differing concentrations of A2E (Figure 1Cf,i,l). These data are consistent with reports in the literature implying that A2E is endocytosed by RPE cells and then concentrated in lysosomal storage bodies. In the cytoplasm, these A2E-containing lysosomal bodies initiate important early inflammatory factors such as interleukin (IL)-1β, followed by the activation of IL-6, IL-8, and monocyte chemoattractant protein-1 (MCP-1). These events occur via the NLRP3 inflammasome, which activates caspase-1; fragmentation of DNA; and deformation of cell membranes, which is associated with the formation of geography atrophy (i.e., advanced dry-AMD) [23,24]. Moreover, some reports have shown that low-concentrations of A2E (10 to 14 nM) induce lysosomal alkalinization in RPE cells, leading to melanogenesis—melanocytes protect RPE cells from oxidative stress [25]. As a result, the controversial role of A2E in RPE cells should be further investigated.

Notably, the association between blue light hazards and chronic retinal degeneration is an important issue that needs to be emphasized, attributable to the widespread usage of electronic products such as smartphones, tablets, and computers [26]. Short wavelength blue light (455 nm to 495 nm) is characterized as high-energy radiation in the visible spectrum and is easily transmitted to the lens, directly causing damage to the retina [27]. Furthermore, blue light reportedly inhibits the activity of superoxide dismutase and catalase [28], and induces RPE cell and photoreceptor death [29]. Previously, we demonstrated that low-luminance BLL induces retinal damage using in vitro and in vivo models. We found that low-luminance BLL regulates apoptotic signals, including the Bax/Bcl-2 and Fas/FasL pathways, and increases protein accumulation of cleaved caspase-3 [30]. Owing to the biological properties of A2E (e.g., photodegradation and oxidation), in the present study we applied these models to A2E-loaded RPE cells and BN rats to estimate the effects of low-luminance BLL and A2E accumulation. In Figure 2e,l, cleaved caspase-3 activity was detected in A2E-laden RPE cells with BLL exposure, which is consistent with reports indicating that BLL-induced photo-toxicity originates from A2E in RPE cells [31]. During illumination of A2E with BLL, reverse-phase high performance liquid chromatography and fast atom bombardment mass spectrometry revealed that the conjugated carbon-carbon chains of A2E and A2E-derived species absorb oxygen molecules, forming a singlet-oxygen quencher. This photochemical change in the structure of A2E involves photo-oxidation, resulting in the formation of a hazardous agent [32].

RPE cells form a hexagonal epithelium located in between the choroidal and retina layers and are vital to the BRB, which prevents substances or xenobiotics from entering the eye [33]. RPE-related BRB dysfunction is a factor in the pathology of both dry- and wet-AMD, as mediated by VEGF [34]. To determine the effects of BLL on RPE barrier integrity, we treated RPE cells with different concentrations of A2E over time and obtained TEER measurements (Figure 3A). When RPE cells were treated with A2E for 3 h, TEER was lower than that of controls, without significant differences. Our interest in this phenomenon led us to increase the incubation time of A2E to 48 h. When RPE cells were treated with A2E from 3 to 48 h (Figure 3A), we found that TEER decreased at 6 h in the control and A2E groups, but recovered after 6 h. This might be attributable to the electric current causing transient increments in membrane permeability during TEER measurements [35]. However, unlike in the control group, TEER did not recover in the A2E groups, showing that A2E potentially alters cell–cell resistance and injures RPE cells. These results might correlate to the binding of A2E and phosphatidylethanolamine (PE) on the membrane, forming A2E–PE adducts that facilitate A2E insertion in the bilayer, causing membrane solubilization [36,37]. Bavik et al. demonstrated that emixustat, a visual cycle isomerase (RPE65) inhibitor, which inhibits the excessive production of lipofuscin and A2E in an ABCA4^-/-^ mouse model, preserving retinal integrity [38].

Next, we wanted to investigate the influence of BLL exposure on the integrity of A2E-laden RPE cells. Under light exposure, A2E is very photosensitive and easily generates reactive oxygen species (ROS) (i.e., singlet oxygen, oxygen anion, and hydroxyl radical) causing cellular damage and death [39]. In Figure 3C, we show discontinued ZO-1-positive strands in A2E + BLL treated rats, which is consistent with reports in the literature suggesting that light exposure causes photo-toxicity via the Rho/ROCK pathway and activates inflammatory factors such as MCP-1 and C-C motif chemokine 11 (CCL11) [40]. As a result, irradiation with BLL enhances A2E photo-oxidation and damage to cellular integrity in A2E-laden RPE cells.

Importantly, BRB breakdown induces retinal swelling and cystoid edema, which are manifested in exudative AMD correlating with complicated factors such as VEGF, platelet-derived growth factor (PDGF), protein kinase C beta (PKC-β), transforming growth factor beta (TGF-β), angiopoietin-2 (Ang-2), etc. [17]. To further this knowledge, we screened the expression of numerous genes via RT-PCR to evaluate BLL-induced inflammation/angiogenesis in RPE cells. Here, *CCL-2*, *GLUT-1*, *IL-1β*, and *MMP-9* expression was significantly up-regulated after exposure to BLL for 2 h (Figure 4A,B). CCL-2, an MCP-1, is a direct mediator of angiogenesis with the ability to induce endothelial migration and proliferation [41]. Furthermore, CCL-2 is generated by Müller cells and involves the infiltration of microglia/monocytes and photoreceptor death in light-induced retinal degeneration [42]. GLUT-1 is characterized as a glucose transporter involved in the promotion of cell proliferation, thought to be mediated via hypoxia-inducible factor (HIF)-1α/von Hippel-Lindau (vHL) and epidermal growth factor (EGF) pathways [43,44]. In addition, other genes are also up-regulated after UV irradiation, including *NGF*, *TrkA*, *artemin*, *bradykinin-1 receptor*, *COX-2,* and *CCL-3* [45], informing the underlying mechanisms of BLL-mediated physiology and pathology.

Previously, we demonstrated that long-term low-luminance BLL induces retinal damage in both in vitro and in vivo models. We found that BLL causes fundus damage, total retinal thickness decrement, photoreceptor atrophy, and neuron transduction dysfunction in a BN rat model [30]. Meanwhile, A2E is a major component in lipofuscin, which is a well-known metabolic deposit between the choroidal and RPE layers [2]. There are currently no effective treatments for dry-AMD, and many unknown mechanisms regarding BLL/A2E-induced photo-toxicity need to be elucidated before pre-clinical animal models can be established. To this end, we applied BLL to A2E-treated BN rats and investigated in detail the toxic phenomena in this animal model. We found that intravitreous injections of A2E cause minor retinal angiogenesis (Figure 6Ae,h,k, yellow arrows), which is consistent with reports implicating that A2E accumulation induces neovascularization via the retinoic acid receptor (RAR) pathway, and up-regulates VEGF expression after laser photocoagulation in BN rats [46], further damaging retinal neuron transduction (Figure 7). The toxic effects associated with photobleaching and A2E oxidation in a *Rdh8*^-/-^*Rdh12*^-/-^*Abca4*^-/-^ mouse model were demonstrated via slow clearance of all-*trans*-retinal-induced A2E accumulation and facilitation of retinal damage after light exposure (10,000 lux) [47]. 

## 4. Materials and Methods

### 4.1. Cell Culture

The cultured RPE cells (ARPE-19 and ATCC^®^ CRL-2302™) were purchased from the American Type Culture Collection (Manassas, VA, USA) and cultured in Dulbecco’s modified Eagle’s medium (Gibco, Grand Island, NY, USA) supplemented with 10% fetal bovine serum in a humidified atmosphere of 5% CO_2_ at 37 °C. RPE cells were cultured on 10-cm cell culture dishes (Orange Scientific, Braine-l’Alleud, Belgium) and the medium was replaced every 2 to 3 days. Upon passaging with 0.05% trypsin-ethylenediamine tetraacetic acid (Gibco), the cells were replated at a 1:4 ratio.

### 4.2. Synthesis of A2E and Treatment of RPE Cells

Chemical products for the synthesis of A2E, including retinaldehyde and ethanolamine, were purchased from Sigma-Aldrich (St. Louis, MO, USA). A2E was prepared and synthesized as previously described [48,49] and stored as a stock solution (1 mM) in dimethyl sulfoxide (DMSO) at −80 °C in the dark. RPE cells were seeded onto 48-well plates for the MTT assay (48-well flat bottom cell culture plate, Orange Scientific, Braine-l’Alleud, Belgium), fibronectin pre-coated transwell inserts (pore size, 0.4 μm) (Corning, Cambridge, MA, USA), 6-cm dishes for immunofluorescence, RT-PCR and Western blot, and 10-cm dishes for TEER assay. All medium changes and treatments were processed under dim light (30–50 lux) in the culture room.

### 4.3. In Vitro Exposure Protocols

For the in vitro studies, we placed two electric LED plates (LED plate, Zami Studio, Changhua, Taiwan) and a total of 360 blue LEDs (12V blue/red LED, Zami Studio, Changhua, Taiwan) in a cell culture incubator. The distance between the light source and cell plates was 5 cm. Luminance was measure via a light meter (LM-81LX, Lutron Electronic Enterprise, Taipei, Taiwan). Blue LED-emitting lights peaked at 460 nm and 150 lux for the indicated time (0 to 12 h) depending on the experimental design.

### 4.4. Cell Viability Assay

Cell viability was determined by the 3-(4,5-dimethyl-2-thiazolyl)-2,5- diphenyl-2H-tetrazolium bromide (MTT) assay. After treating RPE cells with A2E or BLL, MTT (40 μL/well, 0.4 mg/mL) was added and cells were incubated at 37 °C for an additional 40 min. The Formazan product was dissolved in 200 μL DMSO. Absorption was measured at 570 nm by using a microplate reader (MRX-TC; Dynex Technologies, Chantilly, VA, USA). Values were corrected for background absorbance by subtracting the appropriate blanks. Data are from at least six independent assays performed with three replicates each.

### 4.5. Immunofluorescence

Cells were cultured directly on glass coverslips, washed with phosphate-buffered saline (PBS), fixed with 4% paraformaldehyde in PBS for 15 min, permeabilized with 0.2% Triton X-100 in PBS for 15 min, and blocked with 5% FBS in PBS for 30 min. We used anti-cleaved caspase-3 antibodies (Cell Signaling, Danvers, MA, USA) as the first antibody at a ratio of 1:250 for immunofluorescent staining. After incubating for 24 h with glass coverslips, we used rabbit IgG antibodies (DyLight 594) (DI-1594, Vector Laboratories, Burlingame, CA, USA) as the secondary antibody at a ratio of 1:200. Nuclei were stained with 4′-6-diamidino-2-phenylindole (DAPI; AAT Bioquest Sunnyvale, CA, USA), and cells were observed using a Laser CS SP5 confocal spectral microscope imaging system (Leica, Teban Gardens, Singapore).

### 4.6. Reverse-Transcriptase Polymerase Chain Reaction (RT-PCR)

Total RNA was isolated by Tripure Reagent (Roche, Indianapolis, IN, USA) following the manufacturer’s protocol. To analyze mRNA expression of apoptosis-related genes, RT-PCR was performed using glyceraldehyde 3-phosphate dehydrogenase (*GAPDH*) as an internal control. First-strand cDNA was synthesized from 5 μg total RNA at 42 °C for 50 min. The cDNA was amplified in a DNA thermal cycler (MJ Research, Watertown, MA, USA) using the following program: denaturation for 6 min at 95 °C, denaturation for 28 cycles at 35 s and 95 °C, annealing for 35 s at 54 °C, and extension for 45 s at 72 °C, with a final 7-min extension step at 72 °C. PCR products were separated by electrophoresis on 2% agarose gels (PB1200, Bioman Scientific, Taipei, Taiwan) and visualized via ethidium bromide staining. The PCR primers used in this study are presented in Appendix A.

### 4.7. Trans-Epithelial Electrical Resistance (TEER) Measurements in RPE Cells

For these experiments, we used an EVOM2 (World Precision Instruments; Sarasota, FL, USA) epithelial voltohmmeter, which is designed to detect cell–cell resistance. Before measuring TEER, we cultured RPE cells onto the transwell inserts until they reached 100% confluence. Medium from the inserts was replaced after 24 h with fresh medium containing 5, 10, and 30 μM A2E, and cells were then incubated for the indicated times (3, 6, 12, 24, and 48 h). TEER was initially measured after 3 h of treatment, after which the transwells were returned to the incubator until the subsequent time point. We terminated the experiment after 48 h of A2E treatment; the normalized TEER values were corrected for background resistance using the culture-insert and medium and expressed as Ω*cm^2^.

### 4.8. VEGF Quantification

VEGF secreted into the culture medium was measured by means of an ELISA kit (R&D Systems, Minneapolis, MN, USA). RPE cells were plated in 6-cm dishes and cultured to 70–80% confluence. The growth medium was replaced with serum-free medium and cultures were treated as indicated in the text. VEGF present in 1-mL aliquots of culture medium was quantified at the indicated concentration of A2E (0, 10, and 30 μM), according to the manufacturer’s instructions. Results were normalized to the amount of protein per mL.

### 4.9. Animal Handling

BN rats (250–300 g body weight) were purchased from the National Laboratory Animal Center (Taipei, Taiwan) and kept for 30 days under controlled conditions consisting of 12 h/12 h light/dark cycles, at 25 ± 1 °C, 34–48% relative humidity, and *ad libitum* water and food. All experimental procedures involving the use of animals complied with the Association for Research in Vision and Ophthalmology (ARVO) statements for the use of animals in ophthalmic and vision experimental research. The detailed animal experimental protocol below was reviewed and approved by the Institutional Animal Care and Use Committee of Taipei Medical University (approval number: LAC-2016-0442; date of approval: 01-08-2017).

### 4.10. In Vivo Exposure Protocols

An animal exposure chamber designed by our lab, customized from acrylic sheets (Sunpoint Scientific Instrument, Taipei, Taiwan) per our previous study [30], was used for these experiments. The exposure chamber comprised an outer white acrylic box and an inner transparent acrylic box. BN rats were divided into four groups (control group: kept in the animal colony in a dark room while the other experimental groups were exposed to BLL; BLL-exposed group: rats received 3 h BLL exposure per day; A2E-treated group: rats received an intravitreous injection of A2E [2 μL, 30 μM] into the right eye; A2E + BLL co-treated group: rats received an intravitreous injection of A2E [2 μL, 30 μM] into the right eye plus BLL exposure for 3 h per day) and maintained in the dark to avoid pupil constriction. Sixteen rats were divided into four groups: control group (N = 4), BLL group (N = 4), A2E group (N = 4), and A2E + BLL co-treated group (N = 4), and applied to fundus images, fundus angiography, and SD-OCT experiments. For ERG recording, 38 rats were divided into four groups: control group (N = 8), BLL group (N = 10), A2E group (N = 10) and A2E + BLL group (N = 10). Rats were exposed alone, with one rat per exposure chamber for the duration of the experiment, and were not allowed to sleep during the exposure period. The exposure chamber illumination was 150 lux and was measured by a light meter (LM-81LX, Lutron Electronic Enterprise, Taipei, Taiwan). After completing the experiment, all rats were returned to the animal colonies in a cyclic light/dark (250 lux, 12 h/12 h) environment.

### 4.11. Fundus Images and Fundus Angiography Experiments

BN rats were anesthetized using an intramuscular injection of ketamine (65 mg/kg) and xylazine (15 mg/kg) per a previously published protocol [50,51]. Vibrissae were trimmed with scissors to prevent them from interfering with the experiments. Before optical examination, 0.125% atropine sulfate was applied to both pupils (Sinphar, Yilan, Taiwan). Eye angle and position were adjusted to study different parts of the fundus and were covered with 2% Methocel gel (OmniVision, SA, Neuhausen, Switzerland). Image focusing was achieved by moving the rats and the animal holder. Fundus images and fluorescein angiography images were captured with a Micron III retinal imaging microscope (Phoenix Research Laboratories, Tempe, AZ, USA). For fluorescein angiography experiments, 15 mg/kg sodium fluorescein was injected intravenously and images were captured after 30 s.

### 4.12. Spectral-Domain Optical Coherence Tomography (SD-OCT)

After the fundus images and fundus angiography images were taken, we transferred the rats to the SD-OCT system (Phoenix Research Laboratories, Pleasanton, CA, USA), which is customized for retinal imaging of rats or mice. Following the manufacturer’s instructions, we used the A-scan mode to focus on the rat retinas at a 60° to 90° angle according to the conditions of the cornea and OCT data, which were collected from inner retinal images. InSight XL software (Phoenix Research Laboratories) developed by Voxleron LLC (Pleasanton, CA, USA) was used for SD-OCT data analysis.

### 4.13. Electroretinographic (ERG) Recording

A MP36 4-channel data acquisition system (Biopac Systems, Pershore, UK) was connected to a photic stimulator (model ps33-plus; Grass Technologies, West Warwick, RI, USA) to collect ERG recordings. BN rats were kept in a darkroom overnight to eliminate the effects of light exposure from their previous environment. The rats were then prepared for ERG recording. After anesthesia with ketamine and xylazine, rat pupils were dilated with 0.125% atropine sulfate for 10 min. The cornea was covered with 2% Methocel gel (OmniVision, SA, Neuhausen, Switzerland) before recording via diode transistor logic fiber electrodes to increase electronic conductibility and sensitivity. ERG signals were amplified (DC to 300 Hz) and digitized at 1 kHz with a resolution of 2 μV. The ERG recordings were taken from the right eye. A single trace was obtained a total of four times from each rat. The time interval between trace records was 30 s. Multiple waves were taken and the results of 4 trace records were averaged for each animal.

### 4.14. Data and Statistical Analysis

We performed in vivo studies to investigate the retinal damage induced by periodic BLL exposure, we applied the distinction in three layers as previously proposed [50]. For in vitro and in vivo results, data were analyzed by one-way analysis of variance (ANOVA) test. The Student–Newman–Keuls test was used to evaluate statistically significant differences between groups. A *p* < 0.05 was considered to be statistically significant.

## 5. Conclusions

These findings are indicative of BLL-mediated mechanisms involving oxidation and photosensitization of A2E induced by inflammation/angiogenesis in RPE cells, as well as retinal damage in BN rats. Furthermore, these results provide potential biomarkers for further investigation.

## Figures and Tables

**Figure 1 ijms-20-01799-f001:**
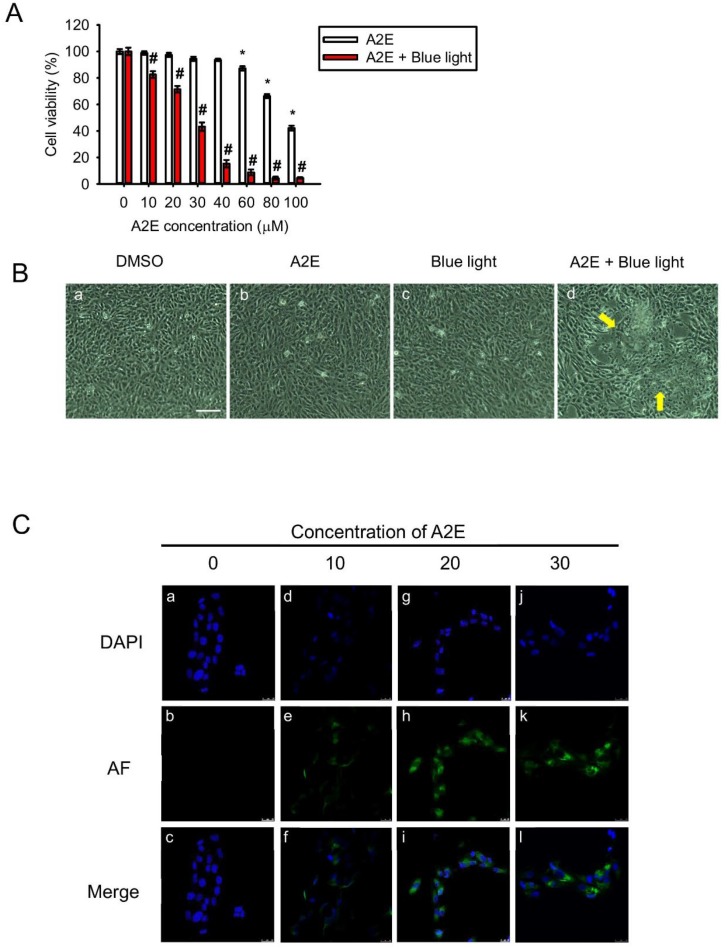
Blue light damages A2E-laden retinal pigment epithelial (RPE) cells. (**A**) Cell viability (%) was measured using the 3-(4,5-dimethyl-2-thiazolyl)-2,5- diphenyl-2H-tetrazolium bromide (MTT) assay. A2E-laden RPE cells with or without blue light-emitting diode (LED) light (BLL) treatments were compared with control cells in the dark; 6 independent experiments were performed with three each. White bars indicate RPE cells exposed to 0, 10, 20, 30, 40, 60, 80, and 100 μM synthetic A2E for 12 h. Red bars indicate RPE cells co-treated with BLL for 12 h (460 nm, 150 lux) and the indicated concentration of synthetic A2E (0, 10, 20, 30, 40, 60, 80, and 100 μM). Data represent the average percent cell viability ± SE. * *p* < 0.05 was considered statistically significant compared to controls. ^#^
*p* < 0.05 compared with the A2E group. (**B**) Morphology of A2E-laden RPE cells with or without BLL exposure is shown. (**a**) A representative figure of RPE cells grown to confluence and loaded with dimethyl sulfoxide (DMSO)as a vehicle control. (**b**) A representative figure of RPE cells grown to confluence and loaded with 30 μM A2E. (**c**) A representative figure of RPE cells exposed to BLL for 12 h. (**d**) A representative image of RPE cells co-treated with 30 μM A2E and exposed to BLL for 12 h is shown. RPE cells were observed to have a slim and shrinking morphology (indicated by yellow arrows). Results from 3 independent experiments are shown (N = 3). Scale bar, 100 μm. (**C**) Photographs showing A2E-laden RPE cell autofluorescence are presented. RPE cells were loaded with the indicated concentration of A2E (10, 20, and 30 μM) for 12 h. Immunofluorescence showing 4′-6-diamidino-2-phenylindole (DAPI) staining (blue) in the cell nuclei and localized autofluorescence (green) in A2E cells is presented. Results from 5 independent experiments are shown (N = 5). Scale bar, 25 μm.

**Figure 2 ijms-20-01799-f002:**
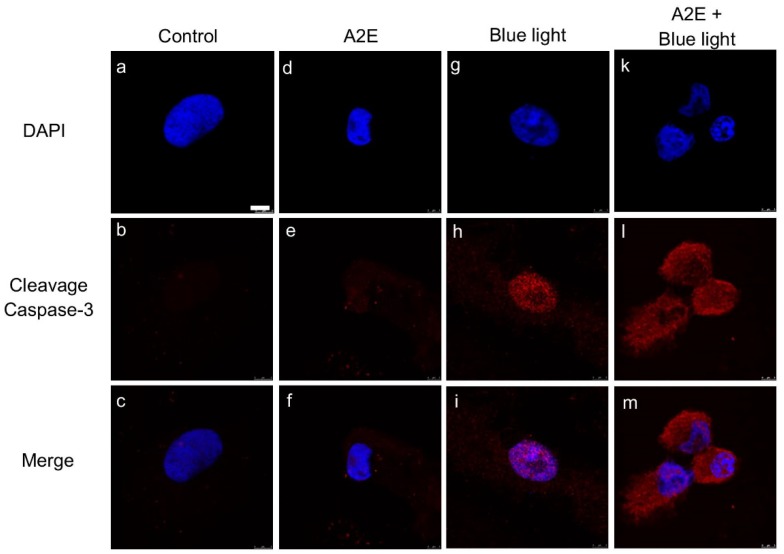
Blue light causes accumulation of cleaved caspase-3 in nuclei of A2E-laden retinal pigment epithelial (RPE) cells. Immunofluorescent staining showed accumulation of cleaved caspase-3 in A2E-laden RPE cells. (**a**–**c**) RPE cells were maintained in the dark and treated with vehicle (dimethyl sulfoxide [DMSO]) as controls. (**d**–**f**) RPE cells were maintained in the dark and treated with 30 μM A2E for 12 h. (**g**–**i**) RPE cells were exposed to blue light-emitting diode (LED) light (BLL) for 12 h (460 nm, 150 lux). (**k**–**m**) RPE cells were co-treated with 30 μM A2E and BLL for 12 h (460 nm, 150 lux). Results from 3 independent experiments are shown (N = 3). Scale bar, 5 μm.

**Figure 3 ijms-20-01799-f003:**
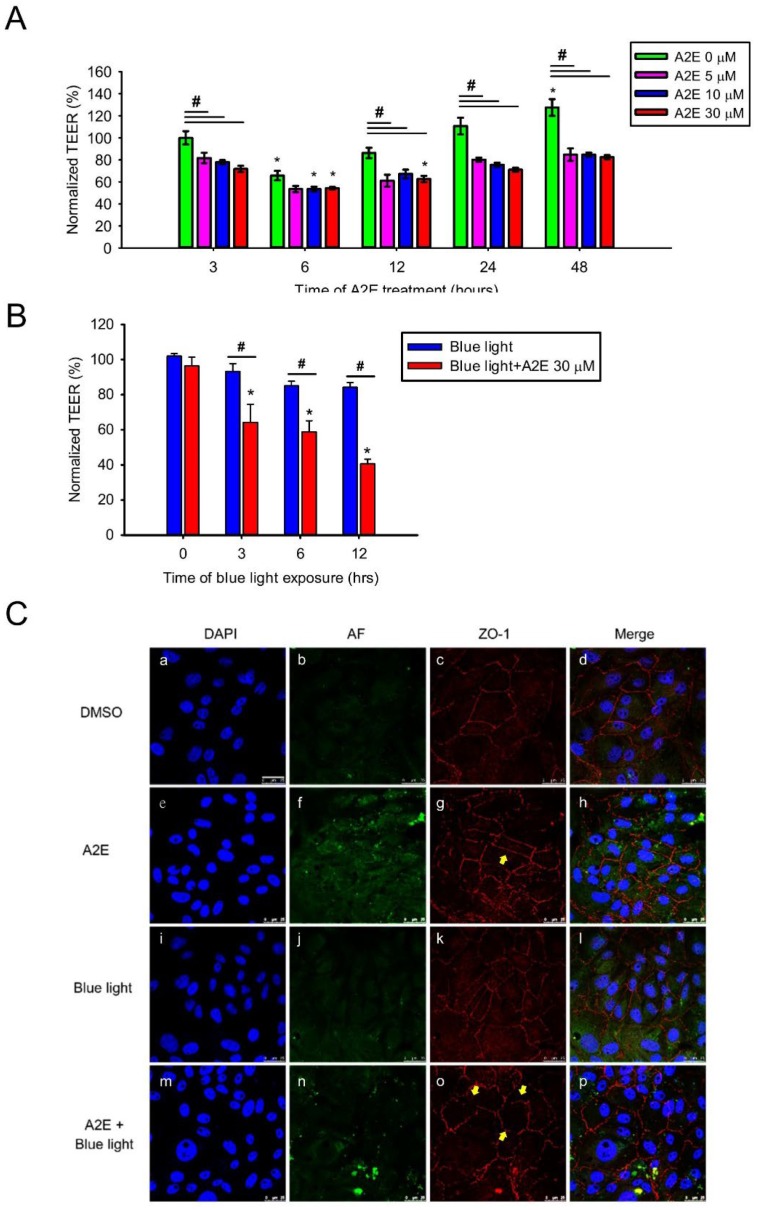
A2E treatment increases paracellular permeability and blue light diminishes epithelial tight junction integrity in A2E-laden RPE cells. (**A**) The effects of osmotic stress on retinal epithelial barrier function are shown. The vertical bars show changes in resistance (Ω*cm^2^) of A2E-loaded RPE cells via trans-epithelial electrical resistance (TEER)measurements. RPE cells were treated with different concentrations of A2E (0, 5, 10, and 30 μM) for the indicated incubation time (3, 6, 12, 24, and 48 h). Results from 4 independent experiments are shown (N = 4). Data are presented as the normalized percent of average TEER ± SE. * *p* < 0.05 compared with the control group treated with same concentration of A2E; ^#^
*p* < 0.05 compared with the 0 μM A2E group during the same treatment time. (**B**) RPE cells were treated with 30 μM A2E and co-treated with different time of blue light-emitting diode (LED) light (BLL) exposure (460 nm, 150 lux; 0, 3, 6 and 12 h). Data are presented as the normalized percent of average TEER ± SE. * *p* < 0.05 compared with the control group treated with 0 h BLL exposure; ^#^
*p* < 0.05 compared with the BLL group. (**C**) Photographs showing A2E autofluorescence (green) and ZO-1 immunofluorescence staining (red) are depicted. Cell nuclei were stained with 4′-6-diamidino-2-phenylindole (DAPI, blue). RPE cells were treated with 30 μM A2E and subjected to BLL for 12 h (460 nm, 150 lux). BLL exposure resulted in the formation of discontinuous ZO-1-positive strands in A2E-laden RPE cells (as indicated by yellow arrows). Results from 3 independent experiments are shown (N = 3). Scale bar, 25 μm.

**Figure 4 ijms-20-01799-f004:**
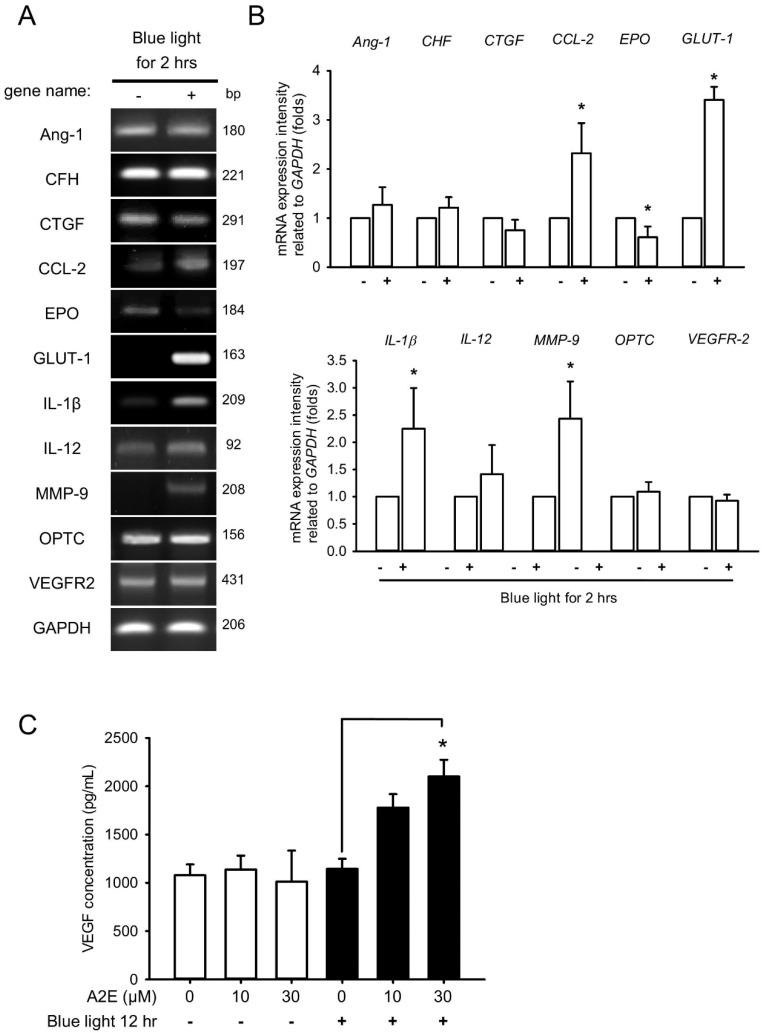
Blue light induces inflammatory and angiogenic gene expression in early stages and increases vascular endothelial growth factor (VEGF) secretion in A2E-laden retinal pigment epithelial RPE cells. (**A**) RPE cells were pretreated with 30 μM A2E for 12 h and then exposed to blue light-emitting diode (LED) light (BLL) for 0 and 2 h (460 nm, 150 lux). Angiopoietin-1 (*Ang-1*), complement factor H (*CFH*), interleukin 1 beta (*IL-1β*), *IL-12*, connective tissue growth factor (*CTGF*), C-C motif chemokine ligand 2 (*CCL-2*), erythropoietin (*EPO*), glucose transporter 1 (*GLUT-1*)*,* matrix metalloproteinase-9 (*MMP-9*), opticin (*OPTC*), VEGF receptor 2 (*VEGFR-2*), and glyceraldehyde 3-phosphate dehydrogenase (*GADPH*) mRNA levels were detected by RT-PCR. (**B**) Quantification of mRNA levels relative to the *GAPDH* control was performed via densitometry and expressed as means ± SE. Results from 3 independent experiments are shown (N = 3). * *p* < 0.05 was considered statistically significant compared to controls. (**C**) The level of VEGF secretion in A2E-laden RPE cells was quantified via enzyme-linked immunoassay (ELISA). RPE cells were loaded with 0, 10, and 30 μM A2E for 12 h (white bars) or co-treated with A2E and blue light-emitting diode (LED) light (BLL) for 12 h (460 nm, 150 lux) (black bars). VEGF secretion increased in A2E-laden RPE cells after BLL exposure. Results from 4 independent experiments are shown (N = 4). * *p* < 0.05 was considered statistically significant compared to controls.

**Figure 5 ijms-20-01799-f005:**
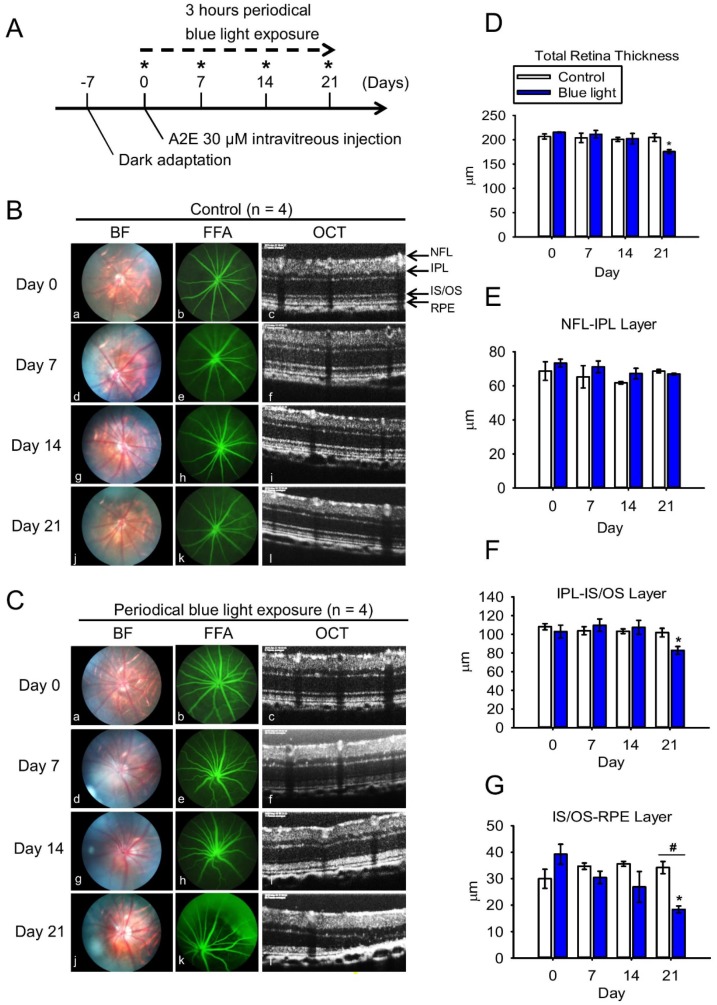
Blue light exposure causes retinal damage in brown Norway (BN) rats. (**A**) The experimental and timeline design are shown. BN rats were divided into four groups: control group, BLL-exposed group, A2E-treated group, and A2E + BLL co-treated group (N = 4 per group). On day 0, day 7, day 14, and day 21 (star marks), BN rats were subjected to the animal ophthalmoscope to obtain fundus images in the bright field (BF), fundus fluorescein angiography (FFA), spectral-domain optical coherence tomography (SD-OCT), and electroretinographic (ERG) recordings. (**B**,**C**) Ocular images from the control group and BLL-exposed group from left to right represent the results of bright field (BF), fundus fluorescein angiography (FFA), and SD-OCT on days 0, 7, 14, and 21. Fluorescein angiography was performed immediately after intravenous injection of 15 mg/kg sodium fluorescein for 30 s. (**D**–**G**). Quantification of data includes total retinal thickness, nerve fiber layer-inner plexiform layer (NFL-IPL), IPL-inner segment/outer segment (IS/OS) layer, and IS/OS-retinal pigment epithelial (RPE) layer in B (white bars) and C (blue bars). Results from 4 independent experiments are shown (N = 4). * *p* < 0.05 was considered statistically significant compared to day 0. ^#^
*p* < 0.05 was considered statistically significant compared to control on the same day.

**Figure 6 ijms-20-01799-f006:**
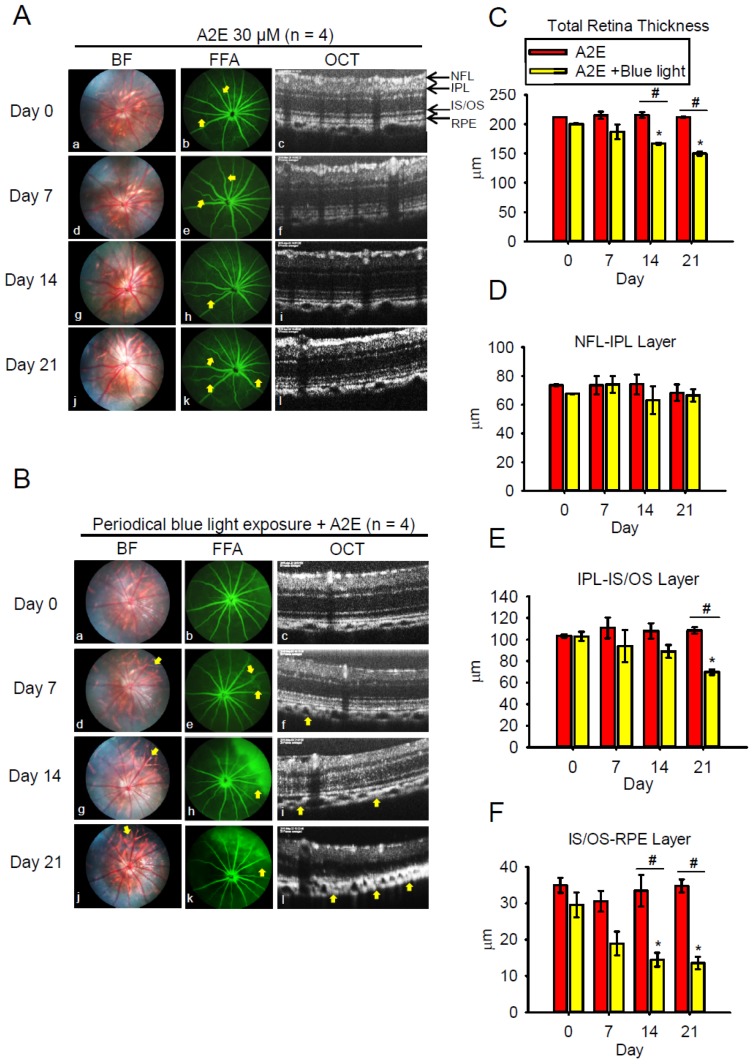
Blue light exposure causes retinal damage in A2E-treated brown Norway (BN) rats. (**A**,**B**) Representative ocular images of the A2E-treated group and A2E + blue light-emitting diode (LED) light (BLL) co-treated group from left to right showing the results of bright field (BF), fundus fluorescein angiography (FFA), and spectral-domain optical coherence tomography (SD-OCT) on days 0, 7, 14, and 21. Fluorescein angiography was performed immediately after intravenous injection of 15 mg/kg sodium fluorescein for 30 s. (**C**–**F**) Quantification of data including total retinal thickness, nerve fiber layer-inner plexiform layer (NFL-IPL), IPL-inner segment/outer segment (IS/OS) layer, and IS/OS-retinal pigment epithelial (RPE) layer in A (red bars) and B (yellow bars). Results from 4 independent experiments are shown (N = 4). * *p* < 0.05 was considered statistically significant compared to day 0. ^#^
*p* < 0.05 was considered statistically significant compared to the A2E-treated group on the same day.

**Figure 7 ijms-20-01799-f007:**
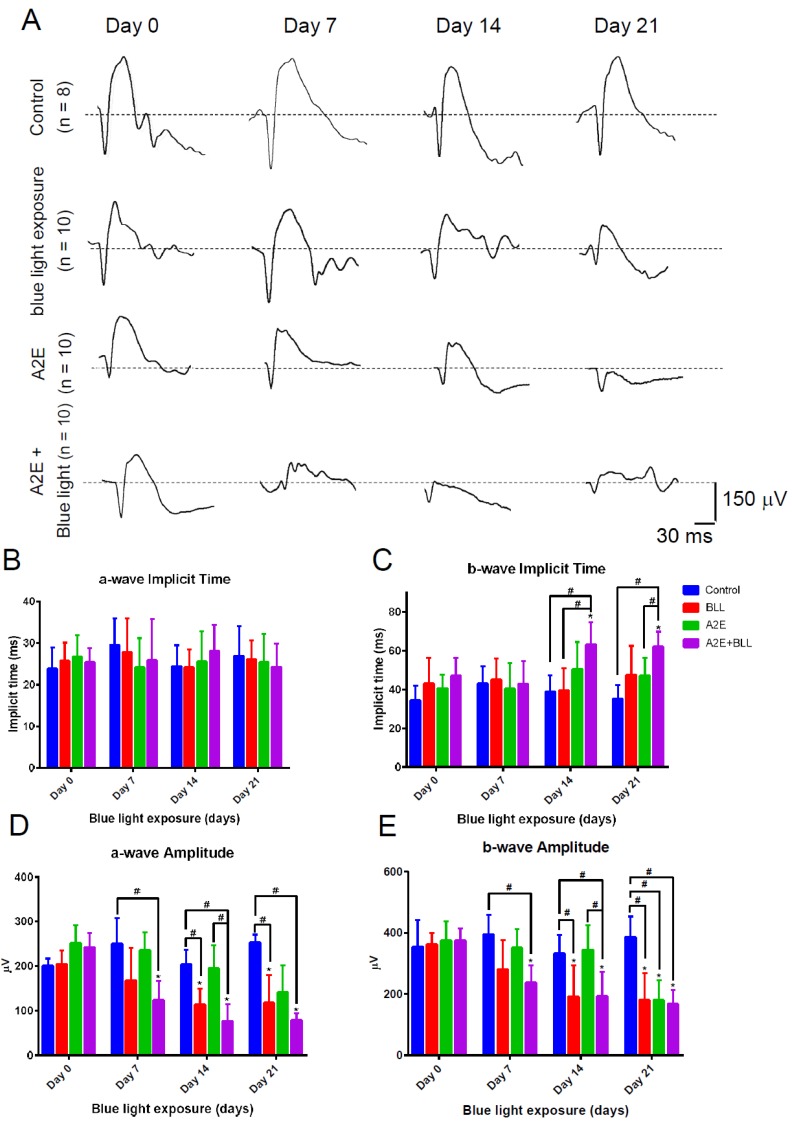
Blue light reduces retinal neuron transduction in A2E-treated Brown Norway (BN) rats. (**A**) Representative electroretinography (ERG) a- and b-waves of the control group (N = 8), BLL group (N = 10), A2E group (N = 10), and A2E + BLL group (N = 10). (**B**–**E**) ERG data were measured on day 0, day 7, day 14, and day 21 for the four groups, and the a- and b-wave amplitude (μV) and implicit time (ms) of the ERG waveforms were analyzed. * *p* < 0.05 was considered statistically significant compared to day 0. ^#^
*p* < 0.05 was considered statistically significant between groups in the same day.

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
