# Peer review of "Low-Luminance Blue Light-Enhanced Phototoxicity in A2E-Laden RPE Cell Cultures and Rats"

_ijms, 2019, doi:10.3390/ijms20071799_

Round 1

Reviewer 1 Report

This study demonstrated the effect of blue light-emitting diode (LED) light (BLL) on phototoxicity in A2E-laden RPE cells and BN rats. They concluded that synergistic effects of BLL and A2E accumulation in the retina increase the risk of retinal degeneration. 

1. In order to demonstrate the effect of BLL in A2E-laden RPE cells, a control group with only BLL is needed. In most of the figures, control groups (DMSO control or BLL control) are missing. 

Fig1A. DMSO control and BLL control are missing. 

Fig1B. BLL group is missing. Yellow arrow is missing.

Fig 3A. BLL control and A2E+BLL group are missing.

Fig 3C.  DMSO control and BLL control are missing.

Fig 4A.  DMSO control and BLL control are missing.

2. The detailed condition of cultured RPE cell should be provided. Highly differentiated polarized monolayers of RPE cells should form a uniform hexagonal lattice that was similar to native RPE tissue in vivo, and is more resistant to cell injury. It’s hard to see the hexagonal shape of RPE cells in the figures. 

3. Although it was mentioned that angiogenesis was induced by BLL or A2E. It’s not clear to see the angiogenesis induced by periodic BLL exposure on day 7 or day 14 by fluorescein angiography.Similarly, It’s not clear to see the angiogenesis induced by A2E on day 7 or 14 or 21. But instead, there is an increase of vessel tortuosity treated with A2E. 

4. The ERG amplitude is not comparable in Fig6 and Fig8. For example, the baseline b-wave is about 350 μv in control group in Fig6, meanwhile the baseline b-wave is about 550 μv in the A2E group in Fig8. It’s better to put data from all four groups into the same figures to compare. 

5. Fig 4C. Why do you chose 2h BLL in Fig 4A and 12h BLL in Fig4C?

6. Fig 5. Please add the label as indicated in the result part.

7. Which part of retina do you choose to measure the thickness by OCT?  How many locations do you measure in the same retina?

8. Please label the “n” in Fig5~Fig8.

Author Response

Responses to reviewers’ comments

Reviewer #1: 

Comments to the Author

This study demonstrated the effect of blue light-emitting diode (LED) light (BLL) on phototoxicity in A2E-laden RPE cells and BN rats. They concluded that synergistic effects of BLL and A2E accumulation in the retina increase the risk of retinal degeneration.

1. In order to demonstrate the effect of BLL in A2E-laden RPE cells, a control group with only BLL is needed. In most of the figures, control groups (DMSO control or BLL control) are missing. Fig1A. DMSO control and BLL control are missing. Fig1B. BLL group is missing. Yellow arrow is missing. Fig 3A. BLL control and A2E+BLL group are missing. Fig 3C. DMSO control and BLL control are missing. Fig 4A. DMSO control and BLL control are missing.

RESPONSE: Thank you for the precious suggestion. In Figure 1A, we used DMSO (marked as 0 μM A2E, white bar) as the DMSO control and 12-hour blue light exposure (marked as 0 μM A2E, red bar) as the BLL control. In Figure 1B, we added BLL group and the yellow arrows as follow. We added BLL control and A2E+BLL group in Fig 3B. In Figure 3C, we added BLL control and A2E+BLL group as follow. In Figure 4A, we didn’t apply A2E in this test so DMSO control is not needed; BLL control in this test is labeled as “Blue light for 2 hrs (-)” in Figure 4A.

  Figure 1B 

  Figure 3B 

  Figure 3C 

2. The detailed condition of cultured RPE cell should be provided. Highly differentiated polarized monolayers of RPE cells should form a uniform hexagonal lattice that was similar to native RPE tissue in vivo, and is more resistant to cell injury. It’s hard to see the hexagonal shape of RPE cells in the figures.

RESPONSE: Followed by your opinion, we revised the description as follow: The cultured RPE cells (ARPE-19 and ATCC® CRL-2302™) were purchased from the American Type Culture Collection (Manassas, VA) and cultured in Dulbecco’s modified Eagle’s medium (Gibco, Grand Island, NY) supplemented with 10% fetal bovine serum in a humidified atmosphere of 5% CO2 at 37°C. RPE cells were cultured on 10-cm cell culture dishes (Orange Scientific, Braine-l’Alleud, Belgium) and the medium was replaced every 2 to 3 days. Upon passaging with 0.05% trypsin-ethylenediamine tetraacetic acid (Gibco), the cells were replated at a 1:4 ratio. Thank you for the suggestion.

3. Although it was mentioned that angiogenesis was induced by BLL or A2E. It’s not clear to see the angiogenesis induced by periodic BLL exposure on day 7 or day 14 by fluorescein angiography. Similarly, It’s not clear to see the angiogenesis induced by A2E on day 7 or 14 or 21. But instead, there is an increase of vessel tortuosity treated with A2E.

RESPONSE: Thank you for the kind suggestion. Followed by your opinion, we revised the description as follow:

Original version: “Further, periodic BLL exposure induced retinal angiogenesis on day 7 (Figure 5Ce), which was even more pronounced on day 14 (Figure 5Ch), as shown by fluorescein angiography.” (page 19, line 10)

Revised version: Further, periodic BLL exposure induced vessel tortuosity on day 7 (Figure 5Ce) and day 14 (Figure 5Ch), as shown by fluorescein angiography.” (page 19, line 10)

Original version: “Though retinal angiogenesis was observed after periodic BLL exposure on days 7 and 14, …” (page 19, line 17)

Revised version: Though retinal vessel tortuosity was observed after periodic BLL exposure on days 7 and 14, …” (page 19, line 17)

Original version: “A2E induced retinal angiogenesis on day 7 (Figure 7Ae), which was even more pronounced on days 14 and 21 (Figure 7Ah and 7Ak).” (page 20, line 11)

Revised version: “A2E induced retinal vessel tortuosity on day 7 (Figure 7Ae), which was even more pronounced on days 14 and 21 (Figure 7Ah and 7Ak).” (page 20, line 11)

4. The ERG amplitude is not comparable in Fig6 and Fig8. For example, the baseline b-wave is about 350 μv in control group in Fig6, meanwhile the baseline b-wave is about 550 μv in the A2E group in Fig8. It’s better to put data from all four groups into the same figures to compare.

RESPONSE: Followed by your opinion, we put ERG data from all groups into the same figure as follow. ERG data were collected on day 0, day 7, day 14, and day 21 for four groups, and the a- and b-wave amplitude (μV) and implicit time (ms) of the ERG waveforms were analyzed. For these results, statistically significant differences were evaluated by using Student–Newman–Keuls test. *p < 0.05 was considered statistically significant compared to day 0. #p < 0.05 was considered statistically significant compared to control group on the same day.

5. Fig 4C. Why do you choose 2h BLL in Fig 4A and 12h BLL in Fig4C?

RESPONSE: In our unpublish data, we found the results regarding to the change of VEGF transcriptional expression with different time of blue light exposure to ARPE cells as follow. The transcription activities of VEGF-A, VEGF-C and VEGF-D started after 4 hours of blue light exposure and reached maximum expression after 12 hours of blue light exposure which provided an important clue of VEGF-related angiogenic effects induced in early stage (before 4-hour of blue light irradiation). Based on this result, we chose 2 hours as the early time point of blue light exposure in Figure 4A and 12 hours as the time point in Figure 4C to test the VEGF protein production.

6. Fig 5. Please add the label as indicated in the result part.

RESPONSE: Thank you for the suggestion. Followed by your opinion, we add the labeling in the Figure 5 as follow.

7. Which part of retina do you choose to measure the thickness by OCT? How many locations do you measure in the same retina?

RESPONSE: While finishing the orientation of the rats’ eye toward the OCT detective camera head, the optic nerve head would be shown in the center of the monitor. In the following figure, yellow lines represented as our scanning area. We did four-time measurements on the same retina and averaged the data of thickness.

8. Please label the “n” in Fig5~Fig8.

RESPONSE: Thank you for your valuable comment. Followed by your precious opinion, we added the n number labeling in Figure 5 to 8.

Reviewer 2 Report

An interesting study by Lin et al showing that blue light enhances phototoxicity in A2E-laden RPE cell culture and rats. This is an addition to the authors’ previous study on blue and red light effects on retinal damage.

Although, majority of experiments seem to be well conducted, I have a few outstanding issues.

Why did the authors choose 150 lux for Blue LED treatment? It is important to provide an appropriate dose-response curve for different light intensities as a basis for the decision on what light to use in all the experiments. The dose-response can be done at least for the cell viability experiments.

Further on, why 12h or 3h BLL exposure was chosen? Similarly, after deciding on the light intensity, a dose-response should be provided for different exposure times. Again, this can be done at least in the cell viability assay.

Detailed information on statistical tests used is missing and should be provided, especially where multiple groups are compared appropriate multiple comparison procedures should be applied.

Description of the light stimulus for ERG experiments is missing and needs to be provided.

In the lines 217-218, it is not clear what did the authors mean by “a dose-dependent increase in autofluorescence”. What is the physiological conclusion that is implied in here?

Panels 3A and 3B seem to be showing the same data just presented in different ways, is it necessary? On the other hand, numerical TEER data with BLL treatment in addition to microscopy images in C should be presented in Figure3.

In the panel 5F is there a significant difference for 21 day treatment?

Author Response

Responses to reviewers’ comments

Reviewer #2

Comments to the Author

An interesting study by Lin et al showing that blue light enhances phototoxicity in A2E-laden RPE cell culture and rats. This is an addition to the authors’ previous study on blue and red light effects on retinal damage. Although, majority of experiments seem to be well conducted, I have a few outstanding issues.

1. Why did the authors choose 150 lux for Blue LED treatment? It is important to provide an appropriate dose-response curve for different light intensities as a basis for the decision on what light to use in all the experiments. The dose-response can be done at least for the cell viability experiments.

RESPONSE: In our unpublish data as followed, we have tried the effects of 150 lux and 200 lux blue light exposure on RPE cell viability in preliminary test. Data represented the average percent cell viability ± SE. *p < 0.05 was considered statistically significant compared to their own controls (0-hour blue light exposure). Results from 3 independent experiments are shown (n = 3). We found the cell viability between two groups were similar except of showing difference after 6-hour exposure. However, there was an over-heat condition happened in our apparatus under 200 lux blue light exposure procedure. As a result, we chose 150 lux for blue light treatment.

2. Further on, why 12h or 3h BLL exposure was chosen? Similarly, after deciding on the light intensity, a dose-response should be provided for different exposure times. Again, this can be done at least in the cell viability assay.

RESPONSE: In our preliminary test, we found the results regarding to the change of VEGF transcriptional expression with different time of blue light exposure to RPE cells as follow. The transcription activities of VEGF-A, VEGF-C and VEGF-D started after 4 hours of blue light exposure and reached maximum expression after 12 hours of blue light exposure which provided an important clue of VEGF-related angiogenic effects induced in early stage. Based on this result, we chose 12 hours as the time point in Figure 4C to test the VEGF protein production.

3. Detailed information on statistical tests used is missing and should be provided, especially where multiple groups are compared appropriate multiple comparison procedures should be applied.

RESPONSE: The experimental results are expressed as the mean ± SE from the indicated number of experiments. The results were analyzed using Student–Newman–Keuls test with the Sigma Stat v3.5 software to evaluate statistically significant differences between the groups. A p-value < 0.05 was considered to be statistically significant. Followed by reviewer’s opinion, we revised the description as follow:

Original version: “*p < 0.05 was considered statistically significant when compared to the same concentration of A2E at 3 h. #p < 0.05 was considered statistically significant compared to 0 μM A2E during the same treatment time.” (page 36, line 11)

Revised version: “*p < 0.05 compared with the control group treated with same concentration of A2E; #p < 0.05 compared with the 0 μM A2E group during the same treatment time.” (page 36, line 11)

4. Description of the light stimulus for ERG experiments is missing and needs to be provided.

RESPONSE: Thank you for your valuable comment. Scotopic ERGs were used to evaluate the cone and rod photoreceptor responses to light stimulus. Our ERG systems were composed of acquisition system and a MP-36 4-channel amplifier (Biopac Systems, Inc., Pershore, UK) connected to a PS33-PLUS photic stimulator (Grass Technologies, Warwick, RI USA). Recordings were obtained using 10-ms flash stimuli with an intensity of 19.1 cd · s/m2. The implicit time of a- and b-waves are measured from the stimulus onset. The amplitude of a-wave was recorded as the length from baseline to the negative peak; the amplitude of b-wave was recorded as the length from the trough of the a-wave to the peak of the positive wave

5. In the lines 217-218, it is not clear what did the authors mean by “a dose-dependent increase in autofluorescence”. What is the physiological conclusion that is implied in here?

RESPONSE: A2E is a lipofuscin fluorophore and exhibits autofluorescence1. In Figure 1C, we applied different concentrations of A2E (10 ,20 and 30 μM) to RPE cells. After A2E treatment, we found autofluorescence in cytoplasm of RPE cell through confocal microscopy. Moreover, the level of autofluorescence increased in a dose-dependent manner showing that we delivered A2E to cytoplasm and generated A2E-laden RPE cells as a model to proceed further investigation.

6. Panels 3A and 3B seem to be showing the same data just presented in different ways, is it necessary? On the other hand, numerical TEER data with BLL treatment in addition to microscopy images in C should be presented in Figure3.

RESPONSE: Thank you for your valuable comment. Follow by your opinion, we revised Figure 3 as follow. We deleted the curve-line histogram and added numerical TEER data with BLL treatment in Figure 3B. In Figure 3B, RPE cells were treated with 30 μM A2E and co-treated with different time of blue light-emitting diode (LED) light (BLL) exposure (460 nm, 150 lux; 0, 3, 6 and 12 hours). Data are presented as the normalized percent of average TEER ± SE. *p < 0.05 compared with the control group treated with 0-hour BLL exposure; #p < 0.05 compared with the BLL group. We also added BLL treatment in Figure 3C.

7. In the panel 5F is there a significant difference for 21 day treatment?

RESPONSE: Yes, it is. We revised the figure in Figure 5F as follow. Thank you for your kind suggestion.

1          Sparrow, J. R., Parish, C. A., Hashimoto, M. & Nakanishi, K. A2E, a lipofuscin fluorophore, in human retinal pigmented epithelial cells in culture. Investigative ophthalmology & visual science 40, 2988-2995 (1999).

Round 2

Reviewer 1 Report

This revised manuscript answered most of the questions. Some minor issues still exist as follows:

The control groups were added in most of the figures in the response letters to reviewers. However, the revised manuscript didn’t include the updated figures. The figure number in discussion part is not updated either. A careful double check with the updated figures, figure legend and the text in the whole manuscript is needed.

For the animal study part, the unclear demonstration about “increased angiogenesis” was replaced with “increased vessel tortuosity”.  Is this phenomenon observed in all the animals? I noticed that the N=4, the percentage (or number) of increased tortuosity on d7 and d14 as well as the percentage (or number) of FFA leakage on d21 should be provided. 

All the ERG data from the 4 groups were put together. However, the problem still exists. It was not explained why there is huge difference in the baseline between the 4 groups. The baseline b-wave is about 350 μv in control group in Fig6, meanwhile the baseline b-wave is about 550 μv in the A2E group in Fig8. The statistic analysis should be done with all 4 groups and showed comparison between each two groups. If the huge difference can not be explained, considering the big variation between individual animal in ERG, 8~10 animals’ data is needed to get a comparable baseline in all 4 groups.

Author Response

Responses to reviewers’ comments

Reviewer #1: 

Comments to the Author

This revised manuscript answered most of the questions. Some minor issues still exist as follows:

1. The control groups were added in most of the figures in the response letters to reviewers. However, the revised manuscript didn’t include the updated figures. The figure number in discussion part is not updated either. A careful double check with the updated figures, figure legend and the text in the whole manuscript is needed.

RESPONSE: Thank you for the precious suggestion. Followed by your opinion, we revised the description as follow:

Original version: “…we treated RPE cells with different concentrations of A2E over time and obtained TEER measurements (Figure 3A and 3B).” (page 17, line 467)

Revised version: “…we treated RPE cells with different concentrations of A2E over time and obtained TEER measurements (Figure 3A).” (page 17, line 467)

2. For the animal study part, the unclear demonstration about “increased angiogenesis” was replaced with “increased vessel tortuosity”. Is this phenomenon observed in all the animals? I noticed that the N=4, the percentage (or number) of increased tortuosity on d7 and d14 as well as the percentage (or number) of FFA leakage on d21 should be provided.

RESPONSE: Followed by your opinion, we provided the percentages of vessel tortuosity increment and FFA leakage are presented in the table and all fundus fluorescein angiography (FFA) data as follow.

Percentage of increased   vessel tortuosity (%) (n=4)

d0

d7

d14

d21

Control

0

0

0

0

BLL

0

75

100

100

A2E

0

50

75

100

A2E+BLL

0

0

50

25

Percentage of FFA   leakage (%) (n=4)

d0

d7

d14

d21

Control

0

0

0

0

BLL

0

25

75

100

A2E

0

0

50

50

A2E+BLL

0

50

100

100

3. All the ERG data from the 4 groups were put together. However, the problem still exists. It was not explained why there is huge difference in the baseline between the 4 groups. The baseline b-wave is about 350 μv in control group in Fig6, meanwhile the baseline b-wave is about 550 μv in the A2E group in Fig8. The statistic analysis should be done with all 4 groups and showed comparison between each two groups. If the huge difference can not be explained, considering the big variation between individual animal in ERG, 8~10 animals’ data is needed to get a comparable baseline in all 4 groups.

RESPONSE: Thank you for the kind suggestion. We revised the ERG data by eliminating extreme b-wave data from day 0 to day 21 and performing statistical analysis. Followed by your opinion, we put the data of b-wave together. Results from 4 independent experiments are shown. *p < 0.05 was considered statistically significant compared to day 0. #p < 0.05 was considered statistically significant compared to control group on the same day. Also, we revised the new b-wave data in Figure 6E and Figure 8E in the manuscript.

Reviewer 2 Report

The authors have provided thorough answers to my questions.

Author Response

Responses to reviewers’ comments

Reviewer #2

Comments to the Author

The authors have provided thorough answers to my questions. 

RESPONSE: Thank you for your precious revision.

Round 3

Reviewer 1 Report

1. The total number of the animal used and the number of animals/group in each experiment should be provided in the method.

2. Detailed information on statistical tests used, especially where multiple groups are compared. Appropriate multiple comparison procedures should be applied. It seems that you only compared between two groups instead of among 4 groups with multiple comparison. 

3. How the extreme b-wave data be eliminated for statistical analysis again if  only 4 data collected only from 4 animals? If deleted 2 extreme data, you only have 2 data left for analysis. Due to the big variation in animal itself, a minimum of 8~10 data is needed for multiple comparison in multiple groups. The ERG data is not reliable with less than 4 animals/group.

4. The figures in the response letters are not showed in the final manuscript.

Author Response

Responses to reviewers’ comments

Reviewer #1: 

Comments to the Author

1. The total number of the animal used and the number of animals/group in each experiment should be provided in the method.

RESPONSE: Thank you for your kind comment. We totally used 54 rats in our in vivo experiments. 16 rats were divided into four groups: control group (N=4), BLL group (N=4), A2E group (N=4) and A2E + BLL co-treated group (N=4) and applied to fundus images, fundus angiography and SD-OCT experiments. For ERG recording, 38 rats were divided into four groups: control group (N=8), BLL group (N=10), A2E group (N=10) and A2E + BLL group (N=10). Followed by your suggestion, we added the above information in the Materials and Methods section (page 4, line 165).

2. Detailed information on statistical tests used, especially where multiple groups are compared. Appropriate multiple comparison procedures should be applied. It seems that you only compared between two groups instead of among 4 groups with multiple comparison.

RESPONSE: Followed by your kind suggestion. We revised our detailed information about the statistical test as follow: data were analyzed by one-way analysis of variance (ANOVA) test. The Student-Newman-Keuls test was used to evaluate statistically significant differences between groups. We added the description above in the Materials and Methods section (page 4, line 205). Also, we re-analyzed our data as follow. *p < 0.05 was considered statistically significant compared to day 0. #p < 0.05 was considered statistically significant between groups in the same day.

3. How the extreme b-wave data be eliminated for statistical analysis again if only 4 data collected only from 4 animals? If deleted 2 extreme data, you only have 2 data left for analysis. Due to the big variation in animal itself, a minimum of 8~10 data is needed for multiple comparison in multiple groups. The ERG data is not reliable with less than 4 animals/group.

RESPONSE: Thank you for the precious suggestion. Indeed, ERG data is not reliable with less than 4 animals per group. We’ve added more animals in our ERG experiments and re-analyze the correlation between groups as mentioned in Q1 and Q2.

4. The figures in the response letters are not showed in the final manuscript.

RESPONSE: Followed by your opinion, we’ve revised and rearranged the figures and the descriptions in our Result and Discussion section. Please refer to our revised manuscript.

Round 4

Reviewer 1 Report

The authors did a great job, made all the needed changes. I don't have any further comments on the updated manuscript.